# Insights into the Adsorption Mechanisms of the Antimicrobial Peptide CIDEM-501 on Membrane Models

**DOI:** 10.3390/antibiotics13020167

**Published:** 2024-02-08

**Authors:** Daniel Alpízar-Pedraza, Yessica Roque-Diaz, Hilda Garay-Pérez, Frank Rosenau, Ludger Ständker, Vivian Montero-Alejo

**Affiliations:** 1Biochemistry and Molecular Biology Department, Center for Pharmaceutical Research and Development, Ave. 26 # 1605, Nuevo Vedado, Ciudad de La Habana 10400, Cuba; daniel.alpizar@cidem.cu (D.A.-P.); yessica.roque957@gmail.com (Y.R.-D.); 2Department of Life and Environmental Sciences, Polytechnic University of Marche, Via Brecce Bianche, 12, 60131 Ancona, Italy; 3Peptide Synthesis Group, Center for Genetic Engineering and Biotechnology, Ave. 31 e/158 y 190, Playa, Habana 11600, Cuba; hilda.garay@cigb.edu.cu; 4Institute of Pharmaceutical Biotechnology, Ulm University, Albert-Einstein-Allee 11, 89081 Ulm, Germany; frank.rosenau@uni-ulm.de; 5Core Facility for Functional Peptidomics, Ulm Peptide Pharmaceuticals (U-PEP), Faculty of Medicine, Ulm University, 89081 Ulm, Germany

**Keywords:** antimicrobial peptides, molecular dynamics simulations, panusin, panulirin, β-sheet peptides, β-hairpin peptides

## Abstract

CIDEM-501 is a hybrid antimicrobial peptide rationally designed based on the structure of panusin and panulirin template peptides. The new peptide exhibits significant antibacterial activity against multidrug-resistant pathogens (MIC = 2–4 μM) while conserving no toxicity in human cell lines. We conducted molecular dynamics (MD) simulations using the CHARMM-36 force field to explore the CIDEM-501 adsorption mechanism with different membrane compositions. Several parameters that characterize these interactions were analyzed to elucidate individual residues’ structural and thermodynamic contributions. The membrane models were constructed using CHARMM-GUI, mimicking the bacterial and eukaryotic phospholipid compositions. Molecular dynamics simulations were conducted over 500 ns, showing rapid and highly stable peptide adsorption to bacterial lipids components rather than the zwitterionic eucaryotic model membrane. A predominant peptide orientation was observed in all models dominated by an electric dipole. The peptide remained parallel to the membrane surface with the center loop oriented to the lipids. Our findings shed light on the antibacterial activity of CIDEM-501 on bacterial membranes and yield insights valuable for designing potent antimicrobial peptides targeting multi- and extreme drug-resistant bacteria.

## 1. Introduction

Antimicrobial resistance (AMR) has become an increasing concern for worldwide healthcare. The World Health Organization (WHO) predicted that by 2050, antimicrobial-resistant infections will become the first cause of death, even over cancer [1,2]. However, in light of the 2020 SARS-CoV-2 pandemic and the exacerbated antibiotic prescription during this time, it is expected that this data will be altered, and the number of deaths attributed to AMR is expected to approach a much higher number due to a global change in antibiotic consumption patterns [3]. In addition, the currently available antibiotics do not offer a complete solution against Multidrug Resistance (MDR) and Extensive Drug Resistance (XDR) bacteria, also called superbugs. Particularly in hospitals and communities, where the arsenal of antibiotics is no longer effective in countering bacterial infections, this situation is heading toward a crisis regarding antibiotic resistance [4]. As a result, this issue represents an attractive and essential field to research for new classes of bioactive compounds to slow down this development. In this sense, antimicrobial peptides (AMPs) have been raised as promising candidates against AMR.

AMPs are naturally present in the innate immune system and have broad-spectrum antimicrobial properties aiding in the defense against invading microorganisms such as most Gram-negative and Gram-positive bacteria, fungi, enveloped viruses, and eukaryotic parasites [5]. Since their discovery, AMPs have been found in virtually all species, including bacteria, fungi, insects, amphibians, crustaceans, birds, fish, mammals, and humans [3]. Unlike conventional antibiotics, most AMPs are believed to exert their antimicrobial activity through membrane interaction and disruption rather than through recognition of a single receptor, making resistance less likely to occur [6]. They are peptides with 12 to 50 amino acid residues long, with +2 to +4 as the most abundant net charge. In addition, this cationic character can be enhanced by amidation at the C-terminus of the peptide. The presence of hydrophobic residues confers the amphipathic character and, in many cases, it represents about 50% of the sequence, with Leu being the most frequent. These properties permit the peptide to fold into an amphiphilic structure in three dimensions, often upon contact with membranes [7]. Based on their structure, AMPs are classified into four broad families: α-helical peptides, β-sheet peptides, αβ-mixed peptides, and extended/flexible peptides; this last group includes Pro-rich, Arg-rich, Lys-rich, and His-rich peptides, among others [8,9].

The β-sheet antimicrobial peptides present a well-defined number of β-strands, with relatively few or no helical domains, organized in the common amphipathic pattern. Most of these peptides are constrained by disulfide bonds providing a scaffold to maintain a well-defined three-dimensional structure. The cysteine-containing β-sheet peptides are a highly diverse group of molecules mainly represented by defensins [10,11]. Analysis of cysteine-rich peptides (CRPs) displaying antimicrobial activity shares a common structural element, the so-called γ-core, which is a conserved sequence pattern for β-defensins. The γ-core is a three-dimensional signature composed of two antiparallel β-sheets connected by a short turn region with a central location in many antimicrobial peptides. It is positively charged and amphiphilic, facilitating membrane interactions [12,13,14]. Several studies have highlighted that γ-core determines the defensin’s antimicrobial properties [14,15,16,17].

Panusin and panulirin are two structurally related β-defensine-like peptides isolated from the spiny lobster *Panulirus argus* [18,19,20]. Panulirin has shown strong trypsin-binding capacity as an inhibitor involved in the prophenoloxidase activation system of the innate immune response [20]. On the other hand, panusin is a β-defensin-like peptide showing broad-spectrum and salt-resistant antimicrobial activity, with any hemolytic activity in human erythrocytes [19]. A recently published study shows that the shorter variant of panusin representing the carboxyl terminus (Ct_PaD) showed higher antimicrobial activity against Gram-negative and Gram-positive bacteria than the parent molecule [21]. The Ct_PaD also showed serum stability with t_1/2_ > 600 min values, non-hemolytic activity, and non-cytotoxic activity on MRC-5 cells. Altogether, these results prompt us to consider Ct_PaD as a valuable antimicrobial lead for the rational design of new antimicrobial peptides based on structure [21]. Consequently, among other candidates, CIDEM-501 was chimerically constructed from the carboxyl terminus regions of panusin [21] and the putative P1 region suggested for panulirin [20]. The rationally designed hybrid peptide has shown antimicrobial activity at low concentrations against multi- and extreme-drug-resistant bacteria without cytotoxic effect against eucaryotic cells (Table 1) [22].

In this study, we employed a molecular dynamics simulations approach to investigate the adsorption mechanism of CIDEM-501 onto bacterial and eukaryotic membrane models. Through a comprehensive analysis of parameters characterizing the peptide-membrane interactions, with the aim to unveil the contributions of structural determinants present in CIDEM-501. Our findings lighten the antibacterial efficacy of CIDEM-501 against bacterial membranes, providing crucial insights that are instrumental in the rational design of potent antimicrobial peptides. These insights are particularly relevant in combating multi- and extreme drug-resistant bacterial strains, underscoring the significance of our research in advancing the development of effective therapeutic strategies.

## 2. Results

### 2.1. CIDEM-501 Secondary Structure Study

The peptide generally presents high stability in both systems, aqueous and in the presence of TFE, showing very stable structural arrangements in the CD spectra. The spectra showed a minimum at 190–200 nm, indicating a disordered secondary structure (SS) with an inflection point around 215–218 nm with low ellipticity, which contributes to the β-strand conformation (Figure 1). Furthermore, they show a maximum of low ellipticity at 230 nm, which is characteristic of peptides with disulfide bonds and the contribution of aromatic residue side chains. CIDEM 501 has been structurally modified with a disulfide bond between Cys2-Cys23 to achieve cyclization of the peptide, thereby conferring structure stability.

A more exhaustive analysis of the primary data obtained in the spectra was carried out using the CDPro program [25,26] to accurately estimate the contributions of secondary structures of the peptide (Figure 1). The tabulated results of CDPro confirm the contribution in the content of beta sheets (~36%) and disordered structure (~59%), with favorable RMSD values (<0.3) based on the comparison with 48 soluble model proteins [27]. All together it indicates that CIDEM-501 is a stable peptide that performs little variation in its SS even in the presence of TFE which promotes the formation of intramolecular hydrogen bonds.

### 2.2. Peptide 3D Structure Prediction

All servers were able to correctly predict the disulfide bridge between Cys2-Cys23, rather they returned different conformations. PEP-FOLD returned rich α-helix structures (H), while RoseTTAFold and AlphaFold2 retrieved two structures based on a mixture of β-strands (B) and disordered structures (T) (Figure 2A), which are in accordance with the obtained data from CD. However, to have a more accurate result and select the theoretical structure closer to the experimental, the percentage of secondary structure for each result was calculated using the secondary structure calculator plugging implemented in VMD (VMD-SS) [28]. From this analysis we obtained a relation of H:B:T of 63:7:29, 11:22:66, and 11:44:44 for the predicted structures by PEP-FOLD, RoseTTAFold, and AlphaFold2, respectively. The theoretical and experimental data obtained from CD experiments revealed that the predicted structure by AlphaFold2 is closer to the experimental (Figure 2B).

### 2.3. Simulations of Peptide-Membrane Interactions

The distance from the center of mass (COM) of the peptide to the COM of the membrane and the interaction area of the peptide with the membrane was calculated to evaluate the absorption of CIDEM-501 with the membrane models at a time of 500 ns (Figure 3).

CIDEM-501 showed a higher stability during the simulations for both the bacterial membrane models. The peptide binds to bacterial membranes instantly regardless of the starting position (model 1 or 2, see Section 4.3) and remains attached throughout the simulation. In contrast, for the eukaryotic (Zw) membrane model, the peptide takes longer to bind to the zwitterionic membrane in both models 1 and 2. Nevertheless, the distances observed at the beginning of GN simulations were attributed to the flipping of CIDEM-501 over the membrane (Figure 3A). Thus, the peptide is located closer (average < 0.5 nm) to the negatively charged lipids head phosphate than in Zw membranes (average > 0.5 nm), where it is slightly away (Table 2). In fact, CIDEM-501 is closer to the center of the Gram-positive (GP) membrane than to the center of the Gram-negative (GN) membrane, especially in model 1 where it was able to pass the borderline of the lipid head. These results are also supported by the analysis of the surface area of peptide–membrane interaction (Figure 3B). The peptide showed higher values of interaction area with the bacterial membranes (>1400 A^2^) than eukaryotic ones, since they reach the stability time at 200–500 ns. It shows constant and stable interactions with GN and GP membranes while in the eukaryotic the peptide won and lost interaction during the simulation. All these results suggest that CIDEM-501 settles smaller distances and larger areas of interaction with the bacterial mimetic membrane in the more extended period evaluated than with the eukaryotic membrane model used.

### 2.4. Orientation of CIDEM-501 over the Membranes

In all simulations, the peptide adopts a parallel conformation with respect to the surface membrane (Figure 4). Notably, with the GN membrane, despite both models exhibiting a similar binding orientation, the electric dipole moment of model 2 was oriented orthogonal to the membrane’s normal surface in the first 300 ns where the electric dipole showed high fluctuations, indicating an unstable orientation. However, the vector reoriented toward the membrane, increasing its stability in the last 200 ns (Figure 4A). In contrast, model 1 stabilized its orientation rapidly after undergoing a conformational change from 10 to 50 ns. The binding energy of these models confirms the earlier results, with Gibbs energies of −190.9 kcal/mol for model 1 and −114.75 kcal/mol for model 2. These differences could be associated with variations in the number of interactions and hydrogen bonds during the simulation. Model 1 exhibited higher numbers of both total contacts and hydrogen bonds than model 2, contributing to the greater stability of this binding orientation (Figure 5).

In GP membrane simulations, the binding orientations of CIDEM-501 varied based on the initial orientation. For model 1, the electric dipole moment is oriented parallel to the surface normal and fluctuated back and forth during most of the simulation, indicating an unstable orientation. In contrast, in model 2, the electric dipole moment exhibited a similar orientation to the GN simulations. It is rapidly and highly stabilized during the entire molecular dynamics, indicating a more stable orientation than model 1 (Figure 4B). These results were substantiated by the Gibbs energy of each model, with values of −110.75 kcal/mol and −162.39 kcal/mol for models 1 and 2, respectively.

Interestingly, model 1 displayed a higher number of total interactions than model 2 (Figure 5A). However, the orientation of model 1 led to the loss of hydrogen bonds, resulting in a lower overall stability. Conversely, model 2, despite having fewer contacts, maintained a larger number of hydrogen bonds, contributing to its higher stability compared to model 1. The number of hydrogen bonds is similar to the obtained for model 1 in GN, which presented the highest number of total contacts (Figure 5B) and Gibbs energy.

Finally, for eukaryotic membranes, the peptides did not reach a stable orientation during the entire molecular dynamics. In both models, high fluctuations were observed, indicating an unstable adsorption of the peptides over these membrane models (Figure 4C). Due to this, further analyses were not performed on these models. All the points mentioned above reinforce the previous results that CIDEM-501 is capable of binding to the three studied membrane models but exhibits higher stability on bacterial ones.

### 2.5. Structural Determinants for the CIDEM-501 Membrane Interaction

To identify structural determinants in the peptide–membrane interaction an analysis per residue was performed in the last 50 ns of simulation of model 1 in GN and model 2 in GP. The essential residues were defined as those with the highest number of interactions. This analysis indicates that the majority of interactions are through charged and aromatic residues from the terminal regions and center loop of the peptide in both complexes (Figure 6A).

In GN simulations, Tyr1, Leu8, Tyr10, Trp11, Arg13, Ala14, Arg15, Lys21, Ser22, Cys23, Arg24, Arg25, and Tyr27, present occupancies higher than 100%, indicating that these residues were able to interact with more than one lipid at the same time (Figure 6B). Additionally, Tyr1, Arg13, Ala14, Arg15, Arg24, and Tyr27, presented values equal to or higher than 300% of occupancy. These residues seem to play a key role in stabilizing the peptide–GN complex through nonpolar and polar interactions. For polar residues such as Arg13, Arg14, Arg24, and Arg25, hydrogen bonds represent more than 50% of the total interaction, indicating an essential role in polar interactions. A similar pattern can be observed for Ser12. On the other hand, residues such as Tyr1, Leu8, Tyr10, Trp11, Ala14, Lys21, Ser22, Cys23, and Tyr27 do not present significative hydrogen bond values, which is indicative that these residues could be critical in the stabilization of the complex mainly by nonpolar interactions. An analysis per lipid revealed that most interaction occurred with dioleoylphosphatidylethanolamine (DOPE) followed by tetramyristoylcardiolipin (TMCL). On the contrary, the interactions with dioleoylphosphatidylglycerol (DOPG) were shallow and only with residue Arg25 (Figure 6B).

For the GP membranes, a similar pattern of interactions was observed. Residues Tyr1, Tyr10, Arg13, Ala14, Arg15, Lys21, Ser22, Arg24, Arg25, and Tyr27, kept values of occupancy higher than 100% (Figure 6C). Compared with GN simulations, residues like Tyr1, Cys2, Leu8, and Trp11, showed a drastic reduction in the total of contacts. But, some other residues such as Arg7, Ser12, Thr20, and Ser26 increased their interactions, as well as, there is a slight increase in the number of residues with hydrogen bonds. Similar to GN simulations, polar residues like Arg13, Arg14, Arg24, and Arg25 seem crucial for the complex stabilization by polar interactions. On the other hand, Tyr1, Arg7, Tyr10, Ser12, Ala14, Lys21, Ser22, Cys23, and Tyr27 could be involved in the stabilization through nonpolar interactions. The analysis per lipid showed a high number of interactions with both negatively charged lipids, DOPG, and TMCL. However, the peptide revealed a higher number of interactions with the latter, indicating a preference for this lipid (Figure 6C).

Finally, an analysis of the hydrogen bond formations per residue was performed to better understand the adsorption kinetics of CIDEM-501 on the bacterial membranes. For both simulations, a good pattern of interactions was found to involve the terminal regions and the center loop. Also, some residues from the S1 β-strands are involved, which includes the residues from N-terminus (Tyr1 and CYS2) and C-terminus (Arg24, Arg25, Ser26, and Tyr27) (Figure 7). However, the center loop takes longer to interact (~40–50 ns), consistent with the conformational change observed for the peptide in Figure 3. For GP simulations, a fast and stable interaction since the starting point was observed for the terminal and center loop (Figure 7B). These results also support the orientation analysis represented in Figure 4.

## 3. Discussion

AMPs have become valuable resources to fight the growing problem of antimicrobial resistance. However, naturally occurring AMPs usually face several problems that limit their potentialities as drug candidates: (1) AMPs damage the cell membrane of eukaryotes and cause nondesirable hemolytic effects; (2) rising production costs and technical problems limit their manufacture; (3) their stability and activity are limited at certain environment conditions as pH, presence of iron and specific serum proteins; and (4) AMPs are readily hydrolyzed by proteases [29]. Due to this, the design and optimization of novel AMPs, either by de novo or template-based design, has attracted increasing attention.

CIDEM-501 is a chimeric AMP rationally designed following a template-based methodology. It was designed from the C-terminus of the β-defensin-like antimicrobial peptide panusin. Previous studies have highlighted that the C-terminus of panusin is mainly responsible for its antimicrobial activity [21]. This fact is related to three features: (1) The relative abundance of aromatic residues in the primary sequence (>15%), (2) positive and polar residues asymmetrically distributed to this region, and (3) this region constitutes the γ-core of this defensin-like peptide. Additionally, to increase the positive charge and probably the proteolytic stability of CIDEM-501, one of the putative inhibitory loop (P4-‘P4’) identified in the sequence of panulirin [20] was added to the construction (Figure 8). With this, it is presumed that there are no significant changes in the secondary structure while increasing the amphipathic character of the peptide (Table 1), hypothetically improving the disrupting action upon the bacterial membrane. The final sequences of CIDEM-501 resulted in a 27 residues peptide stabilized by one disulfide bond, a positive charge of 7+, and the C-terminus amidated [22].

CIDEM-501 has been proven to present broad antimicrobial activity against Gram-negative and Gram-positive bacteria without a cytotoxic effect in eukaryote cells similar to its template panusin [22]. However, experimental works present some limitations to exploring specific residues’ contributions to the antibacterial mechanism of CIDEM-501. These methodologies require long times, high amounts of reagents, and specialized equipment to obtain further molecular-level insight. On the contrary, computational approaches have the potential to substantially reduce the work and resources needed for the studies of the mechanism of action of AMPs at the molecular level. Although computational methods still work predictively, they are a powerful tool nowadays, and efforts aim to provide data with increasingly better approximations. Here, we shed light on the adsorption mechanism and the contributions of specific residues to the antimicrobial activity of CIDEM-501 using bacteria and eukaryotic membrane models through all-atom molecular dynamics simulation.

To perform molecular dynamics simulations, it is vital to have the 3D structure of all system components. However, the structure of CIDEM-501, by experimental methods such as NMR and X-ray diffractions, has not been obtained yet. Due to this, the spatial coordinates of CIDEM-501 were determined by combining experimental and theoretical approaches. There are multiple methods to predict the structure of proteins and peptides, such as de novo folding, homology modeling, molecular dynamics (MD) simulations, and deep-learning-based methods [30,31,32]. These computational design studies have been primarily limited to larger proteins. However, peptides usually present high flexibility and exist as an ensemble of conformations in solution. Therefore, compared with the larger and more rigid proteins, determining the conformations of peptides is much more challenging [33]. Some computational peptide structure prediction methods have been developed to model peptides, such as PEP-FOLD, APPTEST, Peplook, and PEPstrMOD [30,33,34,35,36,37,38]. However, these algorithms’ prediction accuracy is still not entirely satisfactory. Due to those above, the coordinates for CIDEM-501 were predicted using three different algorithms (see Section 4).

Interestingly, the three algorithms returned different results with different contributions of secondary structure components. The experimental secondary structure contribution of the peptide in an aqueous medium and with TFE was obtained to select the proper structure. TFE promotes the formation of local interactions, and an ordered secondary structure is possible to obtain, which is possible to predict accurately by circular dichroism experiments. Furthermore, TFE interacts weakly with nonpolar residues, which does not severely alter hydrophobic interactions within peptides. Consequently, TFE promotes stability rather than inducing peptide denaturation [25,26]. This study revealed that CIDEM-501 presented a highly stable structure composed mainly of disordered structures (~59%) and β-strands (~36%) in both environments.

The combination of these methodologies allowed us to determine that the most probable conformation of CIDEM-501 was that predicted by AlphaFold2. Recent works demonstrated that AlphaFold2 can predict accurately α-helical, β-hairpin, disulfide-rich, and cyclic peptides [39,40,41]. CIDEM-501 presented a highly twisted β-hairpin structure formed by two antiparallels β-strands stabilized by a disulfide bond. This structure resembles the template used for the design of CIDEM-501 due to the C-terminus of panusin enclosing the antiparallel β-strands II and III. Also, it is worth highlighting that the primary sequence of CIDEM-501 is highly similar to the pore-forming arenicin, a β-hairpin peptide [42,43]. The structural alignment of arenicin NMR with the theoretical structure of CIDEM-501 showed high closeness in these structures (Figure 9). CIDEM-501 presented RMSD values of 2.391 Å and 3.616 Å with arenicin-2 and arenicin-1, respectively. These results agreed with the structure-activity relationship principle that proteins with similar sequences can adopt similar spatial conformation and often perform similar activity. All above strengthen the predicted model of CIDEM-501.

Molecular dynamics simulations have been extensively used in studying peptide–membrane interactions as a proposed mechanism of action for many antimicrobial peptides (AMPs). All the parameters to perform simulations were taken from previous works with simulations of well-known membrane-interacting peptides like defensin [44,45], α-helical [46,47], or β-hairpin [48,49]. The initial step in the mechanism of action of AMPs involves its adsorption onto the target membrane through electrostatic attraction between the positively charged amino acids of the peptide and the negatively charged lipids heads on the membrane surface [50,51]. The adsorption efficiency of CIDEM-501, after all-atom molecular dynamics was assessed through the distance, area, orientation, and type of bond formation, which characterize the interaction of the peptide with the surface of Gram-positive, Gram-negative, and eukaryotic membrane models. Our simulations showed that CIDEM-501 can bind to the three membrane models. However, it binds more tightly to the surface of the bacterial membranes than eukaryotic ones. This result is reinforced by the peptide showing a higher area of interactions with the former models. This initial interaction is guided by the positive charge and aromatic residues in the terminal and center loop regions. Polar residues such as Arg13, Arg14, Arg24, and Arg25 allow the approximation of the peptide to the surface membrane helped by electrostatic attraction. On the other hand, several studies have emphasized the preference of the aromatics residues for the interfaces between membrane surfaces and the center, allowing them to act as anchors for the AMPs and proteins [52,53,54].

Once the peptide has attached to the surface of the membrane, it must adopt an optimal orientation for preserving and maximizing its functionality. The orientation of CIDEM-501 was determined by assessing the variation in the cosine of its electrical dipole orientation angles. A cosine value of −1 indicates an antiparallel orientation to the surface normal, pointing toward the membrane; a value of 0 signifies a parallel orientation to the surface; and a value of 1 indicates a parallel orientation to the surface normal, pointing away from the membrane [44]. The significance of this dipole moment in predicting protein orientations on charged or hydrophobic surfaces has been emphasized in various studies [44,55,56]. The peptide CIDEM-501 exhibited a parallel orientation to the membrane surface. However, in the eukaryotic model, CIDEM-501 showed pronounced fluctuations, indicating a lack of stable orientation, contrasting with its rapid stabilization in the most stable bacterial models. This result reinforces our previous findings that despite the peptide binding onto the three kinds of membranes, it showed a preference for the bacterial lipid composition.

Moreover, simulations featuring a moment dipole vector oriented toward the membrane exhibited heightened stability, indicating a fundamental role of this vector in governing the peptide’s orientation. This effect was particularly accentuated in the GP models, where a dipole vector aligned antiparallel to the surface normal of the membrane displayed excellent stability compared to its parallel counterpart, despite the later presented higher interaction area. These results are in agreement with previous studies showing that the binding orientation of proteins to negatively charged surfaces was dominated by this vector [44,45,55].

The most stable models revealed extensive peptide interactions through both the central loop and terminal ends. The peptide’s highly twisted structure and the residue composition of these regions notably facilitated these interactions. The kink of the peptide leads to a slight separation between its antiparallel β-strands and the membrane surface, leading to a loss of interaction by these zones. However, oligomerization studies of the arenicin-2 peptide showed that the peptide suffered a dimerization after binding to the anionic membranes. This process provokes a change from a twisted structure to a more planar structure, diminishing the amplitude of intramolecular motions, leading to an amphipathic structure with a hydrophobic β-structural core, and positively charged residues on its edge [42]. Based on the sequence and structure similitude that presents CIDEM-501 with arenicin-2, we sought that CIDEM-501 could undergo similar mechanisms.

The selectivity of CIDEM-501 could be associated with the differences in the lipid composition for each membrane’s model employed. Membranes of prokaryotic and eukaryotic cells differ considerably in lipid composition. This aspect is crucial for the AMP action since it is thought to be the basis of the specificity of antimicrobial peptides toward the target cell [57]. In bacterial membranes, anionic lipids such as DOPG and TMCL allow the electrostatic interactions between the lipid headgroups and the cationic peptide [10]. But, in eukaryotic cells, the presence of zwitterionic DOPC lipids, which are characterized by a positively charged amino group at their outer head regions, disturbs this interaction. Additionally, sterols, such as cholesterol, are also neutral, and they are critical in regulating membrane fluidity, the formation of specific lipid domains, and the antimicrobial peptide’s activity [10]. For instance, previous studies have shown that cholesterol inhibits melittin-induced calcein leakage of PC lipid vesicles, and the extent of inhibition appears to be dependent on the concentration of membrane cholesterol [58,59].

## 4. Materials and Methods

### 4.1. Circular Dichroism Measurements

Circular dichroism (CD) spectra of the CIDEM-501 peptide were recorded using a Jasco J-1500 spectrofluoropolarimeter (Jasco Corp., Tokyo, Japan). The peptide was prepared in ultra-pure water at a final concentration of 50 µM. The peptide was titrated with the growing concentration of 2,2,2-trifluoroethanol 10, 20, 40, 60, and 80%. Spectra was measured at 24 °C using 1.0 nm of bandwidth and a 1.0 mm optical path-length quartz cuvette. Eighteen scans were averaged between 250 nm and 190 nm and corrected for the background signal by subtracting the spectra of the appropriate control samples without protein. The mean residue molar ellipticity, [q], was calculated as
$${\left[\theta \right]}_{MRW}\left(deg.{cm}^{2}.{dmol}^{-1}\right)=\;\frac{\theta \left(mdegree\right)*MRW}{10*c\left(\frac{mg}{mL}\right)*L(cm)}$$

Ref. [60] where, [θ]_MRW_: mean molar ellipticity by residue, MRW: molecular weight of CIDEM-501 divided by the number of residues minus one, L: optical path (1.0 mm), θ: ellipticity, c: concentration.

The raw data obtained was also analyzed by the CDPro software using the CONTIN/LL algorithm with the reference protein sets 7 (SP48 resolved soluble proteins) as models for comparison.

### 4.2. Peptide 3D Model Prediction

The 3D structure of CIDEM-501 was predicted by ab initio modeling using three different algorithms. The first one is based on a Hidden Markov Model-derived Structural Alphabet implemented in the PEP-FOLD server (https://bioserv.rpbs.univ-paris-diderot.fr/services/PEP-FOLD/, accessed on 31 May 2022) [30]. The second uses the RoseTTAFold algorithm from server Robetta, it employs a three-track neural network (https://robetta.bakerlab.org/, accessed on 26 May 2022) [60]. Finally, a deep learning algorithm was implemented in AlphaFold2 from Google Colabs (https://colab.research.google.com/github/sokrypton/ColabFold/, accessed on 30 March 2023) [61]. From each server, the prediction of the disulfide bridge between Cys2–Cys23 was used as the primary selection criterion for proper folding of the predicted structures. Then, the best model was chosen, comparing the percentage of secondary structure of the models with those obtained in the circular dichroism.

### 4.3. System Construction

Different membrane types were modeled, including those of bacterial and mammalian cells. Two types of bacterial membranes, Gram-negative (GN) and Gram-positive (GP), are prepared based on a mixture of dioleoylphosphatidylglycerol (DOPG), dioleoylphosphatidylethanolamine (DOPE), and tetramyristoylcardiolipin (TMCL2). In bacteria, these lipids are the major anionic and zwitterionic species. To model the membranes of GN bacteria, such as *Escherichia coli*, we simulated an 80:8:12 ratio of POPE/POPG/TMCL2, and to model the membranes of GP bacteria, such as *Staphylococcus aureus* we simulated a 60:40 ratio of DOPG/TMCL2 [62,63]. For mammalian membranes (Zw), of which the lipid composition is more straightforward, zwitterionic dioleoylphosphatidylcholine (DOPC) lipids and cholesterol were used in a ratio of 60:40 [62,63]. The membrane models were generated using the input generator from the CHARMM-GUI website (https://www.charmm-gui.org, accessed on 13 November 2022) [64,65,66,67]. Peptide molecules were added to one side of the membrane with its center of mass (COM) at 30 Å from the COM of the membrane, mimicking in vitro experiments in which peptide molecules are initially added to the external monolayer of liposomes.

Additionally, simulations were carried out with two initial poses of CIDEM-501 on the membrane surfaces to obtain proper samplings. In both cases, the peptide is parallel to the membrane. However, in model 1, the electric dipole moment vector is pointed out to the membrane. In model 2, the peptide was rotated 180° around the axis, locating the vector to the membrane surface (Figure 10).

### 4.4. Molecular Dynamics Simulations

All simulations were performed using the NAMD 2.14 package [68] with the CHARMM36 force field [69,70,71]. This force field has proved accurate in reproducing the physic–chemical properties of classic antimicrobial peptides and their interactions with membranes [44,49,72]. This force field has been implemented in the CHARMM-GUI server, a highly versatile tool for building biomolecular systems. The TIP3P water model was used to generate explicit solvation conditions [73] and Newton’s equations of motion were integrated using the Verlet (leapfrog) algorithm [74]. Periodic boundary conditions were applied in all directions, and the short-range van der Waals interaction cutoff was 1.2 nm. The particle mesh Ewald method [75] was applied to treat long-range electrostatic interactions, with a 1.2 nm real-space contribution cutoff for Coulombic interactions. A temperature of 310 K and a pressure of 1 atm were maintained by the Langevin thermostat and barostat, respectively [76,77]. In all systems, the protonation states of peptides were assigned based on calculations at pH 7 and with 150 mM NaCl, as used in the experiments. The systems were equilibrated in two steps. In the first place, a 1000-step minimization followed by 0.5 ns of equilibration with the protein constraint was performed to guide the system to the nearest local energy minimum in configuration space. Secondly, the peptide was released from the harmonic constraints and the whole system was further equilibrated by another 0.5 ns. After the equilibration process, all simulations were performed for 500 ns under an isothermal−isobaric (NPT) ensemble without any restraints.

### 4.5. Orientation Angle

Several studies have characterized the protein orientation on membrane surfaces based on the orientation angle between the surface normal vector and hydrophobic (θ) or electric (ψ) dipole moments of the protein. In this work, the orientation for the electric dipole angle was used to compare the effects of different types of membranes on CIDEM-510 orientations. The electric dipole moment of the protein, μ, is calculated using the *measure dipole* option implemented in the software molecular dynamics simulation 1.9.4a (VMD), as follows:$$\mu =\sum_{i=1}^{N}q_{i}x_{i}$$
where qi is the partial charge of atom i and xi is the position of atom i from the COM of the molecule.

### 4.6. Binding Free Energy Calculations

In order to estimate the relative binding free energy of CIDEM-501 to the different types of bacterial membranes, the MM-GBSA (MM, molecular mechanics; GB, generalized Born; SA, surface area) method was employed [78]. In preparation for the MM-GBSA calculations, snapshots of the system configuration were extracted from the last 50 ns of the MD trajectories, and the explicit water molecules and ions were removed. The MM-GBSA analysis was performed on three subsets of each system: the membrane alone, the peptide alone, and the complex. For each of these subsets, the free energy was calculated as follows:$$G_{tot}=\;H_{MM}+G_{sol-pol}+G_{sol-np}-{T\Delta S}_{conf}$$
where H_MM_, G_solv−pol_, G_solv−np_, and S_conf_ corresponded to the sum of bonded and Lennard–Jones energy terms, the polar contribution of solvation energy, the nonpolar contribution to the solvation energy, and the conformational entropy, respectively. Both H_MM_ and G_solv−pol_ were calculated by using NAMD 2.14 with the generalized Born implicit solvent model and the parameters described above. The dielectric constant of the solvent was set to 78.5. G_solv−np_ was calculated as a linear function of the solvent-accessible surface area (SASA) using the following equation:$$G_{solv-np}=\gamma SASA+b$$
which was calculated with a probe radius of 1.4 Å. The constants γ and b were set to 0.00542 kcal/mol/Å and 0.92 kcal/mol, respectively [79]. The binding free energy of the membrane-peptide complexes was calculated by the equation:$${\Delta G}_{bind}=\;G_{tot}\left(membrane-peptide\right)-G_{tot}\left(membrane\right)-G_{tot}\left(peptide\right)$$
where G_tot_ values were the averages over the simulation.

## 5. Conclusions

The CIDEM-501 model represents a β-hairpin peptide composed of two antiparallel β-strands stabilized by a disulfide bond. Our molecular dynamics simulations have elucidated the peptide’s adsorption mechanism, revealing its ability to rapidly and stably attach to bacterial membrane components, particularly in comparison to eukaryotic model membranes. The peptide is able to recognize the bacterial membranes through electrostatic attraction between the Arginine and the anionic lipids. Once, the peptide binds to the membrane it adopts an optimal orientation parallel to the surface membrane. This orientation is governed by the electric dipole vector and stabilized by aromatic and polar residues. Aromatic residues act as anchors of the peptide in the membrane while polar residues stabilize the complex by polar interactions. The level of theory employed in this work does not allow us to arrive at a conclusive mechanism of action. However, due to the similarities found with arenicin-2 we sought that CIDEM-501 could exert membrane disruption through a toroidal pore model. In order to validate this hypothesis further theoretical and experimental studies are required.

## Figures and Tables

**Figure 1 antibiotics-13-00167-f001:**
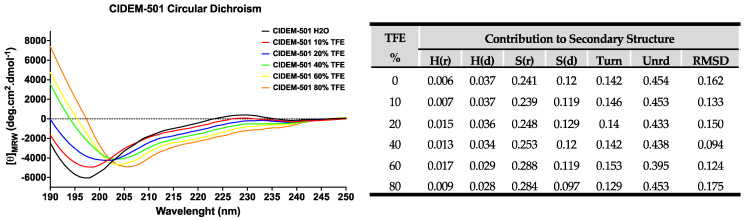
Secondary structure estimation of CIDEM-501 in the presence of TFE. The table shows the results of the SS prediction with CDpro using CONTINN/LL with the reference protein set 7.

**Figure 2 antibiotics-13-00167-f002:**
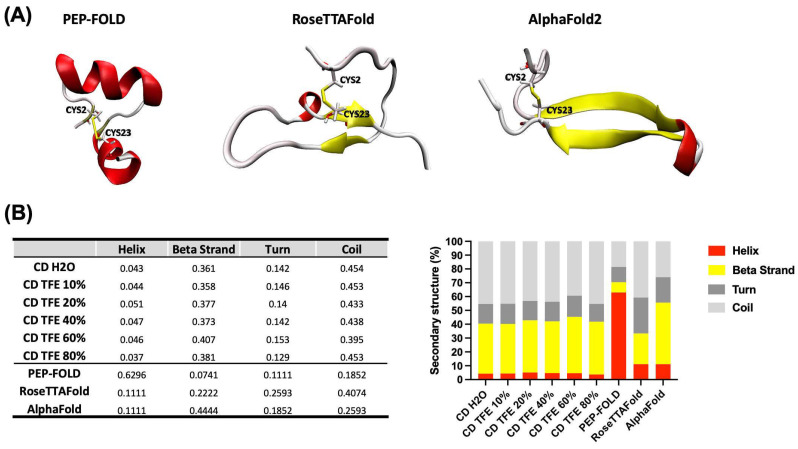
Correlation of theoretical and experimental 3D structure prediction of CIDEM-501. (**A**) 3D structure was obtained using three different servers, PEP-FOLD, RoseTTAFold, and ALphaFold. (**B**) Qualitative and quantitative comparison between theoretical and experimental predictions. In all cases, the colors red, yellow, and gray represent helix, β-strand, and disordered structures, respectively.

**Figure 3 antibiotics-13-00167-f003:**
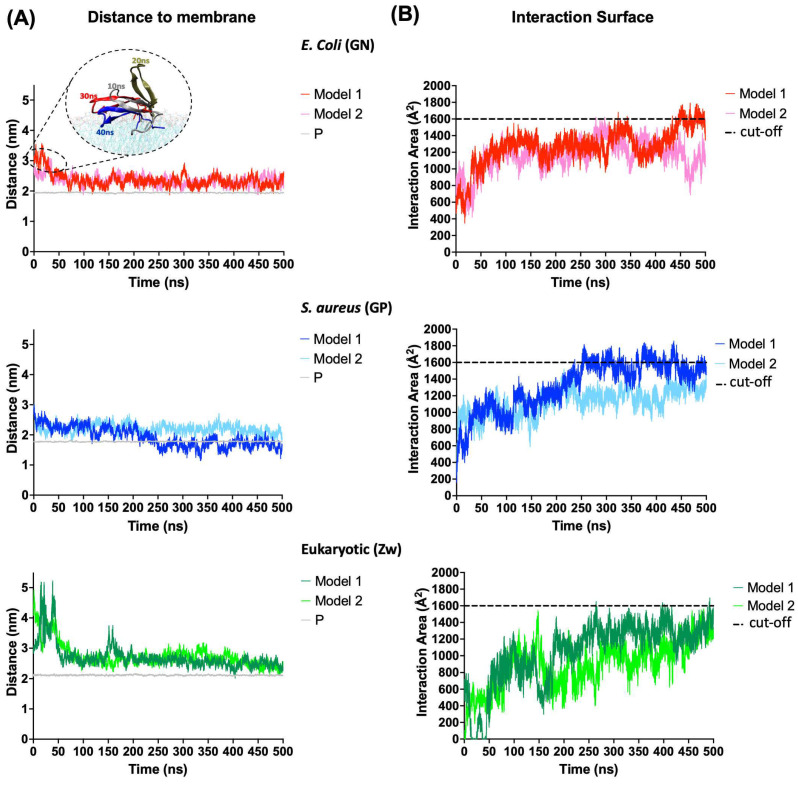
Adsorption of the CIDEM-501 on the membrane’s models. (**A**) Distance from the COM of the peptide to the COM of the membrane. The average distance between COM of phosphorus atoms and COM of the membrane is referred to as P (gray line). (**B**) Interaction area of the peptides with membrane models. Discontinued lines represent a random cut-off to allow an easier comparison.

**Figure 4 antibiotics-13-00167-f004:**
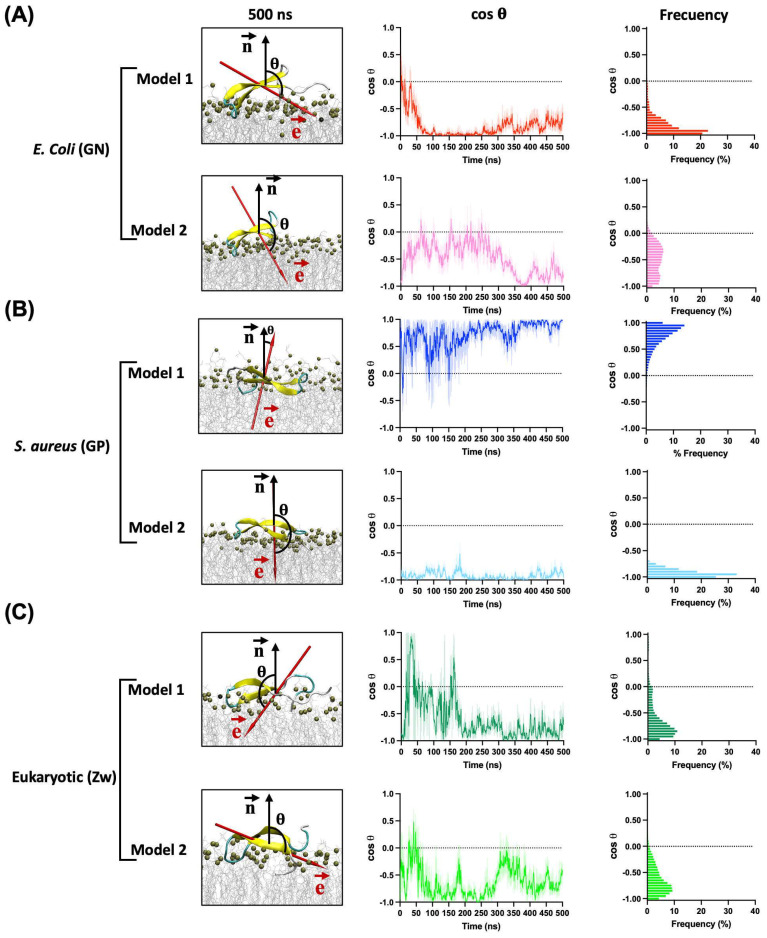
Orientation of CIDEM-501 on the GN (**A**), GP (**B**), and Eukaryotic (**C**) membrane models according to the angle (θ) between the electric dipole ($\vec{e}$, red arrow) and the membrane surface normal ($\vec{n}$, black arrow) in function of time. The final snapshots of each simulation are presented.

**Figure 5 antibiotics-13-00167-f005:**
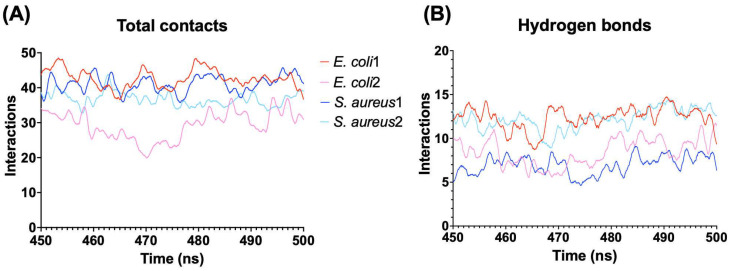
Interaction between the peptides and bacterial membrane models. (**A**) Total of interactions using a distance cutoff of 3 Å from the peptide. (**B**) Number of hydrogen bonds between the peptide and membrane models.

**Figure 6 antibiotics-13-00167-f006:**
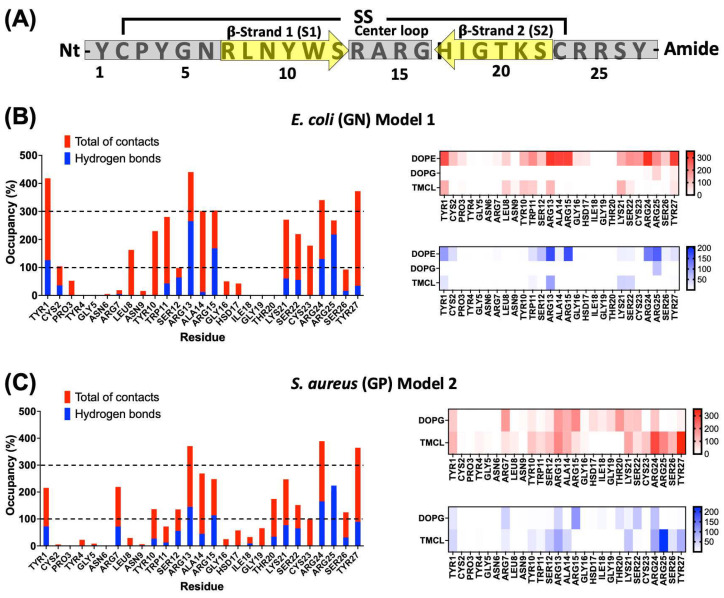
Interaction analysis per residue of the peptide-bacterial membrane complexes. (**A**) General depiction of the relation between primary sequence and secondary structure for CIDEM-501. Gray bars and yellow arrows represent the disordered and β-sheets structures, respectively. Per residue interaction of CIDEM-501 with GN (**B**) and GP (**C**). The histograms represent the total of contacts (red) and hydrogen bonds (blue) during the last 50 ns for each residue. The heat maps include the analysis per residue and lipid. To facilitate analysis, discontinued lines represent two random cut-offs (100% and 300%).

**Figure 7 antibiotics-13-00167-f007:**
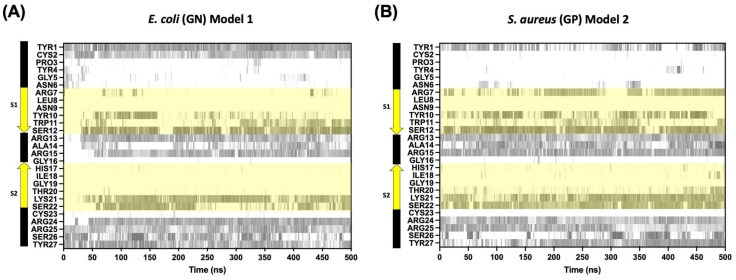
Timeline evolution of the hydrogen bond formation among CIDEM-501 models and Gram-negative membrane composition (**A**) Gram-positive membrane composition and (**B**) the region highlighted in yellow is the residues from β-strands 1 (S1) and 2 (S2).

**Figure 8 antibiotics-13-00167-f008:**
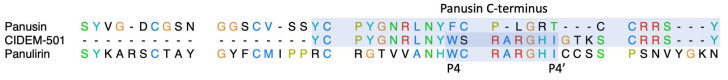
Sequence alignment of Panusin, CIDEM-501, and Panulirin. Residues colors correspond with the properties of conserved residues, blue (hydrophobics), red (positive charged), green (polar), orange (glycines), yellow (prolines), cyan (aromatic), and black (unconserved).

**Figure 9 antibiotics-13-00167-f009:**
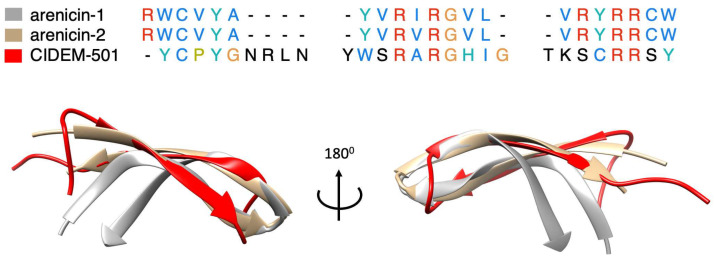
Sequences and structures comparison of CIDEM-501 (red) and the two arenicin isoforms, arenicin-1 (gray) and arenicin-2 (brown). Residues colors correspond with the properties of conserved residues, blue (hydrophobics), red (positive charged), green (polar), orange (glycines), yellow (prolines), cyan (aromatic), and black (unconserved).

**Figure 10 antibiotics-13-00167-f010:**
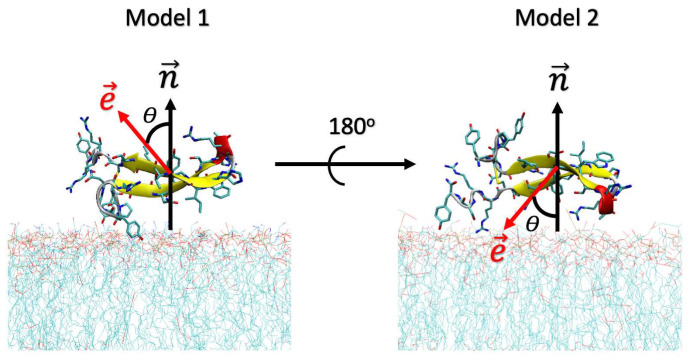
Initial orientations of the peptide CIDEM-501 over the membrane’s surface. The red and black arrows represent the direction of the electric dipole moments ($\vec{e}$) and the surface normal ($\vec{n}$). *θ* represents the angle between both vectors.

**Table 1 antibiotics-13-00167-t001:** Physical-Chemical properties of Panulirin, PaD, Ct_PaD, and CIDEM-501.

ID	Sequence	Cys Pairing	MW (Da) ^a^	Net Charge ^b^	Aliphatic Index ^c^	Hydrophobic Moment ^d^	Boman Index (kcal/mol) ^e^	MIC (μM) ^f^	
								*E. coli*	*S.* *aureus*
Panulirin	SYKARS**C_1_**TAYGYF**C_2_**MIPPR**C_3_**RGTVVANHW**C_4_**RARGHI**C_5_****C_6_**SSPSNVYGKN-amide	C_1_–C_5_ C_2_–C_4_ C_3_–C_6_	5367.2	8+	42.70	0.17	1.86	ND	ND
PaD	SYVGD**C_1_**GSNGGS**C_2_**VSSY**C_3_**PYGNRLNYF**C_4_**PLGRT**C_5_C_6_**RRSY-amide	C_1_–C_5_ C_2_–C_4_ C_3_–C_6_	4260.5	4+	34.87	0.16	1.99	12.5	12.5
Ct_PaD	Y**C_1_**PYGNRLNYF**C_2_**PLGRT**C_3_C_4_**RRSY-amide	C_1_-C_4_ C_2_-C_3_	2801.6	5+	33.91	0.26	2.59	3.1	3.1
CIDEM-501	Y**C_1_**PYGNRLNYWSRARGHIGTKS**C_2_**RRSY-amide	C_1_-C_2_	3260.57	7+	32.59	0.15	3.41	2–4	2–4

^a^ Molecular Wight determined experimentally, ^b^ Net charge (C-terminus amidation included); ^c^ Aliphatic Index determined by the R package Peptides [23], ^d^ Hydrophobic moment determined by R package Peptides [23], ^e^ Boman index represents the binding potential of peptides on bacterial membranes [24], ^f^ Minimum inhibitory concentration determined experimentally [21,22]. ND not determined.

**Table 2 antibiotics-13-00167-t002:** Average of adsorption depth.

Simulations		Depth Adsortion
*E. coli* (GN)	Model 1	0.42
	Model 2	0.41
*S. aureus* (GP)	Model 1	0.14
	Model 2	0.43
Eukaryotic (Zw)	Model 1	0.58
	Model 2	0.6

All units are in nanometers. Adsorption depth was measured as the distance of CIDEM-501 to the phosphate heads.

## Data Availability

The data presented in this study are available on request from the first and corresponding authors.

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
