# Peer review of "Insights into the Adsorption Mechanisms of the Antimicrobial Peptide CIDEM-501 on Membrane Models"

_antibiotics, 2024, doi:10.3390/antibiotics13020167_

Round 1
Reviewer 1 Report
Comments and Suggestions for Authors
1. Can you provide more details about the rationale behind choosing the structure of panusin and panulirin as templates for designing CIDEM-501? Are there specific characteristics in these structures that were particularly influential in the design process?
2. The molecular dynamics simulations focused on the adsorption mechanism of CIDEM-501 to different membrane compositions. What were the key parameters analyzed to understand individual residues' contributions to the interactions, and how do these findings relate to the observed antibacterial activity?
3. The discussion mentions several challenges associated with naturally occurring AMPs, including damage to eukaryotic cell membranes and hemolytic effects. How does CIDEM-501 address or overcome these challenges, especially considering its template-based design?
4. The introduction of the putative inhibitory loop from the trypsin inhibitor panulirin to CIDEM-501 is highlighted in the discussion. Could you discuss the rationale behind incorporating this specific loop and how it contributes to the overall properties of CIDEM-501?
5. The discussion mentions limitations of naturally occurring AMPs, such as hydrolysis by proteases. How does CIDEM-501 address proteolytic stability, and were there specific considerations in the design process to enhance stability under various environmental conditions?
6. The selectivity of CIDEM-501 against bacterial membranes. Can you elaborate on the significance of this selectivity in the context of combating antibiotic resistance, and how it differentiates CIDEM-501 from other antimicrobial peptides?
Author Response
- Can you provide more details about the rationale behind choosing the structure of panusin and panulirin as templates for designing CIDEM-501? Are there specific characteristics in these structures that were particularly influential in the design process?
We thank the reviewer's observation. In the current version of the manuscript, a summary of the basis of the rationale is explained (see lines 85-92). We also consider stating some details to the reviewer. Panusin and Panulirin have been used as templates for the design of CIDEM-501. Both templates (synthetic variants) showed antimicrobial action against Gram-negative and Gram-positive bacteria. The design of CIDEM-501 followed part of the results of structure-activity relationship studies (unpublished). On the other hand, the region described as one of the putative P1 sites for the inhibitory activity of panulirin, was used in the C-terminus variant of panusin with their antimicrobial activity, stability and 3D structure, which have been recently published (DOI: 10.3390/pharmaceutics15061777).
- The molecular dynamics simulations focused on the adsorption mechanism of CIDEM-501 to different membrane compositions. What were the key parameters analyzed to understand individual residues' contributions to the interactions, and how do these findings relate to the observed antibacterial activity?
To understand the contribution of individual residues to the interaction with the membrane, those residues with the higher number of interactions in the last 50 ns of the simulation were evaluated. For this, we employed two methodologies. First, we calculate the total of contacts for each residue using a cut-off = 3 Å, so all atoms closer than 3 Å to a specific residue were considered a contact. Then, we calculate the number of hydrogen bonds for each residue and compare how it represents the total of contacts. Additionally, two cut-offs were established to analyze the plotted data. A first cut-off at 100% occupancy because those residues with higher values than 100% indicate interactions with more than one lipid during the simulation. Also, we included a cut-off of 300% to evaluate those residues more involved in the interaction with the membranes. This analysis identified critical residues for the peptide adsorption onto the bacterial membranes as aromatic (Tyr) and positive charged (Arg) residues. The formers have already been described as essential for the peptide's anchor to the membrane, which is helped by its physic-chemical properties. The arginine helps the peptide bind to the membrane by electrostatic attraction between its positive charge and the negative charge of the phospholipid heads.
Additionally, these residues contribute to the physic-chemical properties of the peptides (described in Table 1, added in this revised version). Table 1 summarizes that CIDEM-501 presents a higher positive net charge, contributing to a strong interaction against negatively charged membranes. Also, the Boman index gives an overall estimate of the potential of a peptide to bind to membranes or other proteins as receptors to normalize; it is divided by the number of residues. A protein has a high binding potential if the index value exceeds 2.48, as was predicted for CIDEM-501, with a Boman index of 3.41.
- The discussion mentions several challenges associated with naturally occurring AMPs, including damage to eukaryotic cell membranes and hemolytic effects. How does CIDEM-501 address or overcome these challenges, especially considering its template-based design?
The authors appreciate the reviewer's interest in the template-based design of CIDEM-501. In this regard, our results in the present research are precisely looking for the explanation of selective toxicity of the new peptides using a theoretical approach.
We designed CIDEM-501, taking into consideration previous studies. Panusin is a naturally occurring AMP that shows no hemolytic activity (DOI: 10.1016/j.dci.2016.09.002). Like other β-defensins, the hemolytic activity showed by pansuin is less than 2% at very high concentrations (10 x MIC). More recently, our group reported the antimicrobial activity of the panusin C-terminus (PaD-Ct), which also showed no hemolytic activity up to 250 µM with the MIC value around five µM (DOI: 10.3390/pharmaceutics15061777). Once we obtained the synthetic variant CIDEM-501, no hemolytic activity was observed till 10 x MIC (not published yet but referred to in the published patent Peptide entities with antimicrobial activity against multi-drug resistant pathogens. Cuba 2020. Inventors: Montero-Alejo V, Pedomo-Morales R, Vázquez-González A, Garay-Pérez H.
- The introduction of the putative inhibitory loop from the trypsin inhibitor panulirin to CIDEM-501 is highlighted in the discussion. Could you discuss the rationale behind incorporating this specific loop and how it contributes to the overall properties of CIDEM-501?
Done. It is included in the revised version (lines 328-333) and Table 1 (line 101)
- The discussion mentions limitations of naturally occurring AMPs, such as hydrolysis by proteases. How does CIDEM-501 address proteolytic stability, and were there specific considerations in the design process to enhance stability under various environmental conditions?
We appreciate the reviewer's comments on proteolytic stability of CIDEM-501. We are still working on this important aspect. The present manuscript addresses the molecular insight of the new peptide CIDEM-501 in its interaction with model membranes. Several drug design approaches have been considered in the studies of the structure-activity relationship of CIDEM-501, such as substituting punctual L-aminoacid by the D-series and the capped N-terminal with alkyl groups; those variants are protected by patent. On the other hand, adding a putative P1 site for trypsin inhibitory activity (resembling panulirin) has yet to be studied with proteases from serum, microorganisms, or other sources. However, our previous work demonstrates that the PaD-Ct showed no proteolytic cleavage when incubated with serum for at least 3 hours (referred to in lines 87-88 of the revised version) (DOI: 10.3390/pharmaceutics15061777).
- The selectivity of CIDEM-501 against bacterial membranes. Can you elaborate on the significance of this selectivity in the context of combating antibiotic resistance, and how it differentiates CIDEM-501 from other antimicrobial peptides?
CIDEM-501 seems active against bacterial membrane integrity by the destabilizing interaction with anionic phospholipids and somewhat, till now, lacks receptor-mediated lytic activity. Thus, acting against supramolecular structures like membranes avoids the classical mechanisms of bacterial resistance to conventional antibiotics, where the basis are specific mutations in target molecules. In this sense, the MIC values of CIDEM-501 against non-resistant and resistant bacterial strains are in the same order of magnitude. Then this can also justify the broad spectrum of antimicrobial activity against Gram-negative and Gram-positive bacteria. Compared with other antimicrobial peptides like polymyxins, it has action mechanisms mediated by the attraction with the negatively charged LPS from the outer membrane of Gram-negative bacteria. Nowadays, the expression of the resistance gene, mcr-1, causes modification of lipid A in the bacterial outer surface, resulting in reduced affinity for polymyxins. The specificity of polymyxins to its targeted LPS probably makes the difference between polymyxins and chimeric peptide CIDEM-501 mechanisms of action demonstrated in the present study. The membrane models we represent in the MS correspond to the inner membrane of Gram-negative and the membrane of Gram-positive bacteria.
Reviewer 2 Report
Comments and Suggestions for Authors
It is appreciable that the authors address the possibility of the effectiveness of putative peptide-type antibiotics to combat increasing infectious diseases. A computational method has been used to study a membrane complex, which may add some value to the field. However, it needs some improvements.
1. Authors claim novel antimicrobial peptides (line 20). It may not be novel because there are several antimicrobial peptides of the same class that have been reported. https://doi.org/10.3389/fmicb.2020.582779
https://doi.org/10.1186/s40779-021-00343-2
2. Abstract section: A value or figure demonstrating the antimicrobial effects of CIDEM-501 is missing, so the abstract is incomplete.
3. More details of CIDEM-501, such as Mol. wt. 3D, amino acid sequence (is it shown in the complete sequence in Figure 6A?) Should have been provided (or a link where this information can be found).
4. What the authors are describing as models 1 and 2 is, in fact not clearly described. It appears that the only difference is in the orientation in which the membrane interacts with it.
5. Figure 3 indicates there is no significant variation in the result of simulation with the prokaryotic membrane (bacteria) to the eukaryotic membrane (host/human), but it needs clarification.
6. There is no mention of protocol optimization and/or validation. Simulation should be compared with any standard antibiotics or known peptides that interact with the membrane of bacteria.
7. The conclusion is lengthy and unclear.
8. Some grammar and typos: Line 2: CIDEM-501 is not a quite common and familiar word and is not to be preferred with this word.Some examples;
Line 39: should be SARS-CoV2.
Line 53: should be Gram-negative
Line 481-483: The name of the bacteria should be in italics.
In Figure 5A: E. coli1 S. aureus1
Comments on the Quality of English Language
8. Some grammar and typos: Line 2: CIDEM-501 is not a quite common and familiar word and is not to be preferred with this word.Some examples;
Line 39: should be SARS-CoV2.
Line 53: should be Gram-negative
Line 481-483: The name of the bacteria should be in italics.
In Figure 5A: E. coli1 S. aureus1
Author Response
It is appreciable that the authors address the possibility of the effectiveness of putative peptide-type antibiotics to combat increasing infectious diseases. A computational method has been used to study a membrane complex, which may add some value to the field. However, it needs some improvements.
The authors appreciate the time dedicated to such revision and thank the reviewer for the comments to improve the quality of this research work.
- Authors claim novel antimicrobial peptides (line 20). It may not be novel because there are several antimicrobial peptides of the same class that have been reported. https://doi.org/10.3389/fmicb.2020.582779
https://doi.org/10.1186/s40779-021-00343-2
Done. Changed the word "novel" for "new" (line 20 of the revised version).
- Abstract section: A value or figure demonstrating the antimicrobial effects of CIDEM-501 is missing, so the abstract is incomplete.
Done. This is introduced in Table 1 (line 101) and also in abstract section (line 21)
- More details of CIDEM-501, such as Mol. wt. 3D, amino acid sequence (is it shown in the complete sequence in Figure 6A?) Should have been provided (or a link where this information can be found).
Done. This is introduced in Table 1 (line 101).
- What the authors are describing as models 1 and 2 is, in fact not clearly described. It appears that the only difference is in the orientation in which the membrane interacts with it.
We are describing models 1 and 2 as the starting positions of the peptides over the membrane. To making more explicit for readers, it was changed the sentence (lines 179-181) where it appears at the very beginning of the results section:
"The peptide binds to bacterial membranes instantly regardless of the starting position (model 1 or 2, see Material and Methods Section 4.3) and remains attached throughout the simulation."
The sentences explain to the reader that we are talking about two different starting positions of the peptide and more information can be obtained in the Materials and Methods section (lines 515-517):
"Additionally, simulations were carried out with two initial poses of CIDEM-501 on the membrane surfaces to obtain proper samplings. In both cases, the peptide is parallel to the membrane. However, in model 1, the electric dipole moment vector is pointed out to the membrane. In model 2, the peptide was rotated 180o around the axis, locating the vector to the membrane surface (Figure 10)."
- Figure 3 indicates there is no significant variation in the result of simulation with the prokaryotic membrane (bacteria) to the eukaryotic membrane (host/human), but it needs clarification.
We performed the depth adsorption analysis in Table 2 because the graphs in Fig. 3 only allow a visual interpretation of the molecular dynamics simulations. Here, we measure the average distance of the COM of the peptide to the COM of the phospholipid heads during all of the MD simulations. The values in Table 2 demonstrate that the peptide binds to the three membranes. Still, the distance is closer for the peptide-membrane interaction in the bacterial models (< 0.5 nm) than for the peptide-membrane interaction in the zwitterionic model (> 0.5 nm).
- There is no mention of protocol optimization and/or validation. Simulation should be compared with any standard antibiotics or known peptides that interact with the membrane of bacteria.
The authors are grateful for the reviewer's comment about the protocol and its validation used in this research work.
For the molecular dynamics protocol was added the following sentences (lines 398-402).
“Molecular dynamics simulations have been extensively used in studying pep-tide-membrane interactions as a proposed mechanism of action for many antimicrobial peptides (AMPs). All the parameters to performed simulations were taken from previous works with simulations of well-known membrane-interacting peptides like defensin [44,45], a-helical [46,47], or b-hairpin [48,49]”
And (lines 526-529)
“This force field has proved accurate in reproducing the physic-chemical properties of classic antimicrobial peptides and their interactions with membranes [44,49,72]. This force field has been implemented in the CHARMM-GUI server, a highly versatile tool for building biomolecular system”
- The conclusion is lengthy and unclear.
We agree with the reviewer's opinion about the conclusions. Lines 464-475 have rearranged as conclusions, providing better understanding and clarity. We hope the reviewer is satisfied with this change, which we believe improved our work.
- Some grammar and typos:
We have corrected the detected errors. However change the name of the peptide “CIDEM-501” would generate confusion referring to published work in the past and in the future.
Line 2: CIDEM-501 is not a quite common and familiar word and is not to be preferred with this word. Some examples;
Line 39: should be SARS-CoV2.
Line 53: should be Gram-negative
Line 481-483: The name of the bacteria should be in italics.
In Figure 5A: E. coli1 S. aureus1

Round 2
Reviewer 2 Report
Comments and Suggestions for Authors
Thank you for the update. Almost all concerns have been addressed.
The conclusion is still lengthy, such as the description of the peptide and methodology, unless it is extremely important.